# Complete remission of diabetes with a transient HDAC inhibitor and insulin in streptozotocin mice

Hideto Kojima [1,2,7 ✉], Miwako Katagi[1,7], Junko Okano[3], Yuki Nakae[2], Natsuko Ohashi[4], Kazunori Fujino[5], Itsuko Miyazawa[6] & Takahiko Nakagawa [1,2 ✉]

Despite the growing epidemic worldwide, diabetes is an incurable disease. We have been focusing on why diabetes manifests refractoriness to any therapy. We recently found that abnormal bone marrow-derived cells (BMDCs), namely, Vcam-1+ST-HSCs, was a key mechanism for diabetic complications. We then hypothesize that those aberrant BMDCs sustainedly impair pancreatic β cells. Here we show that eliminating abnormal BMDCs using bone marrow transplantation results in controlling serum glucose in diabetic mice, in which normoglycemia is sustained even after cessation of insulin therapy. Alternatively, abnormal BMDCs exhibiting epigenetic alterations are treated with an HDAC inhibitor, givinostat, in diabetic mice. As a result, those mice are normoglycemic along with restored insulin secretion even following the cessation of both insulin and givinostat. Diabetic cell fusion between abnormal BMDCs and resident cells is significantly blocked by the combination therapy in the pancreatic islets and thymus while surgical ablation of the thymus completely eliminates therapeutic protection in diabetic mice. In conclusion, diabetes is an epigenetic stem cell disorder with thymic disturbances. The combination may be applied to patients aiming at complete remission from diabetes in clinical medicine.

[1] Department of Biocommunication Development, Shiga University of Medical Science, Otsu, Japan. [2] Department of Regenerative Medicine Development, Shiga University of Medical Science, Otsu, Japan. [3] Department of Plastic and Reconstructive Surgery, Shiga University of Medical Science, Otsu, Japan. [4] Department of Medicine, Division of Diabetology, Endocrinology and Nephrology, Shiga University of Medical Science, Otsu, Japan. [5] Department of Critical and Intensive Care Medicine, Shiga University of Medical Science, Otsu, Japan. [6] Department of Education Center for Medicine and Nursing, Shiga University of Medical Science, Otsu, Japan. [7] These authors contributed equally: Hideto Kojima, Miwako Katagi. ✉email: kojima@belle.shiga-med.ac.jp; takahiko@belle.shiga-med.ac.jp

The growing epidemic of diabetes throughout the world[1] has indicated that better therapy is required to induce the remission of diabetes. Recent studies have put much effort into identifying the causal mechanisms of diabetes. However, those attempts have not typically been linked with eliminating the root of diabetes. Alternatively, we have been interested in why diabetes cannot be cured after disease onset rather than identifying the pathogenesis of diabetes. In other words, we have assumed that a potential mechanism underlies the refractoriness to therapies in diabetes.

The β cells have regenerative potential[2,3], which is predominantly stimulated at the peri-term, peaks at 2 years of age, and is gradually weaned until 5 years old[4]. In contrast, the adult pancreas does not exhibit a proliferative response but retains regenerative potential in response to several stimuli. For example, the β-cell mass could be reversibly expanded and accompanied by an increase in insulin secretion occurs during pregnancy[5]. Another case would be associated with insulin resistance in obesity, in which insulin is increasingly secreted for serum glucose control. Precise mechanisms for a compensatory increase in insulin secretion remain unknown, but animal studies indicate that a compensatory increase in β-cell mass[6], perhaps due to β-cell proliferation, could be a mechanism[7,8]. Alternatively, animal models undergoing pancreatic duct ligation have an expanded β-cell mass[9], although the mechanism for β-cell proliferation remains controversial[10]. Nevertheless, these data indicate that adult β-cells likely retain regenerative potential. However, it has been believed that the regeneration ability of β cells is likely lost under diabetic conditions[11]. There might be machinery for interfering with β-cell regeneration under diabetic conditions.

We have recently found that abnormal bone marrow stem cells (BMDCs) play a key role in developing diabetic complications. These abnormal cells are generated under high glucose conditions and express several inflammatory cytokines, including TNF-α, and importantly, the phenotype of those cells cannot be corrected by serum glucose control using insulin[12]. These cells are also capable of fusing with several types of resident cells to impair their physiological functions[13–18]. BMDCs were finally identified as short-term hematopoietic stem cells (Vcam-1+ST-HSCs)[12]. The causal role of Vcam-1+ST-HSCs was demonstrated by the fact that eliminating these cells alleviated diabetic neuropathy in STZ-induced diabetic mice[12].

We hypothesized that abnormal BMDCs interfere with β-cell regeneration under diabetic conditions and that treating abnormal BMDCs would result in restoring β-cell regenerative potential and increasing the release of insulin from the pancreatic islets. Since abnormal BMDCs cannot be corrected only by glucose control or insulin therapy[12], alternative approaches are required. In the present study, we attempted to examine how β-cell regeneration is interfered with under diabetic conditions using the established diabetic mouse model, the streptozotocin (STZ)-induced diabetes model. This model would be appropriate to address the issue as we assumed that there might be machinery to interfere with β-cell regeneration under diabetic conditions. In other words, if we could remove the unknown causal factors, β cells could regenerate, and diabetes could be brought into remission in this model.

To treat abnormal BMDCs, two steps are likely required; one is to block the de novo production of those cells and the other is to correct epigenetic alterations in abnormal BMDCs. We assume that glucose control at the initial treatment phase would be indispensable to block the additional production of abnormal BMDCs. Therefore, in the present study, we initially utilized insulin to break the vicious cycle of generating abnormal BMDCs. In turn, to treat abnormal BMDCs, we utilized an HDAC inhibitor, as the BMDCs were found to exhibit epigenetic changes.

Finally, we also examined the thymus as this organ is physiologically composed of BMDCs so abnormal BMDCs could alter thymic function, while successful therapy for diabetes could recover thymic function as well.

## Results

**Aberrant BM is the cause of diabetes.** We hypothesized that abnormal BMDCs could disturb the regeneration process of β-cells, and be the reason for the refractoriness to current therapy. If it is the case, replacing abnormal BMDCs with physiological BMDCs could be a basic therapeutic strategy. In addition, we also assumed that glucose control at the initial phase of therapy would be required to block additional production of BMDCs, as abnormal BMDCs are easily produced under high glucose conditions[12]. In turn, insulin would not be required after healthy BMDCs could regenerate β cells in the islets of diabetic mice. To test our hypothesis, diabetes was induced in male C57BL/6J mice by injecting STZ. Insulin or a blank pellet was subcutaneously administered to the diabetic mice. Although the insulin pellet was absorbed over time and faded in a couple of weeks, an additional pellet was never inserted. After 9 Gy of radiation, diabetic mice underwent bone marrow transplantation (BMT) from nondiabetic mice or diabetic mice (Fig. 1a).

Four groups of diabetic mice were prepared to examine the role of BMDCs in diabetic mice (Fig. 1a). Insulin pellets were used to initially control blood glucose in diabetic mice, which were then divided into two groups. One group underwent BMT from nondiabetic mice (yellow line in Fig. 1b), whereas the other received bone marrow from diabetic mice (gray line in Fig. 1b). In the former group, we assumed that the transplanted bone marrow cells from nondiabetic mice would not turn out to be deleterious, as serum glucose would be controlled by insulin while those cells might drive islet regeneration and release endogenous insulin. In the latter group, diabetic mice with bone marrow from diabetic mice might remain diabetic even though serum glucose was controlled by insulin because insulin could not correct the abnormalities in the BMDCs. Therefore, this group would presumably show an increase in serum glucose after the insulin pellets faded away because bone marrow from diabetic mice would not be able to rescue the islets. In the rest of the mice, blank pellets were administered and divided into two groups as controls (blue and orange lines in Fig. 1b).

As shown in Fig. 1b, insulin pellets initially succeeded in lowering serum glucose concentrations in diabetic mice after bone marrow transplantation either from nondiabetic (yellow line: non-DM→DM$^{Ins}$) or diabetic mice (gray line: DM→DM$^{Ins}$). Despite the fade-out of the insulin pellets, serum glucose continued to be normal in the former group (yellow line); however, it increased in the latter group (gray line). The transplanted nondiabetic bone marrow cells contributed to keeping diabetic mice normoglycemic even after the insulin pellet faded, whereas diabetic bone marrow cells failed to do so (Fig. 1b). In turn, blank pellets failed to control blood glucose concentration in diabetic mice reconstituted with bone marrow from either nondiabetic (orange line: non-DM→DM$^{Blk}$) or diabetic mice (blue line: DM→DM$^{Blk}$). Statistical analysis was performed only with diabetic mice with BMT from nondiabetic mice and showed that bone marrow from nondiabetic mice was capable of maintaining significantly lower glucose concentrations at 8 weeks even after the insulin pellet had disappeared (Fig. 1c). Of note, five out of seven diabetic mice receiving bone marrow from diabetic mice (DM→DM$^{Blk}$ and DM→DM$^{Ins}$) were dead by 8 weeks, while no diabetic mice died that were reconstituted with bone marrow from nondiabetic mice (non-DM→DM$^{Blk}$ and non-DM→DM$^{Ins}$), suggesting that bone marrow from diabetic mice was deleterious.

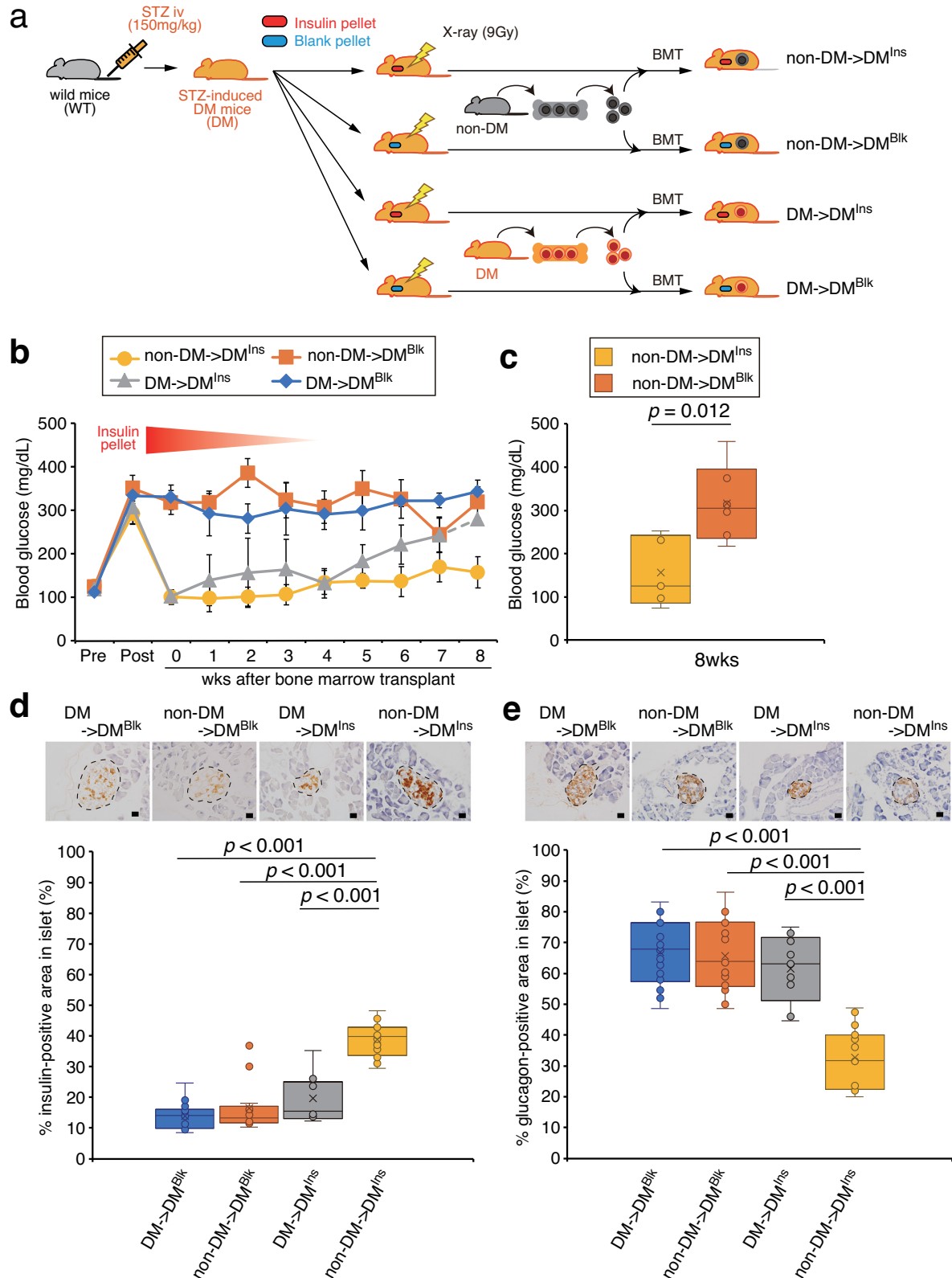

We next examined whether the therapy could recover insulin and glucagon expressions in the pancreas at the end of the experiment. First, STZ-induced diabetic mice were confirmed to exhibit a marked reduction in insulin levels and substantial elevations in glucagon levels in the pancreatic islets compared to those of nondiabetic wild-type mice (Supplementary Fig. 1). Among these four groups, non-DM→DM$^{Ins}$ mice showed significantly higher levels of insulin (yellow bar in Fig. 1d) and lower levels of glucagon in the pancreas (yellow bar in Fig. 1e) at the end of the study compared to those in the other groups. The other groups did not show significant changes in insulin or glucagon levels compared to control DM→DM$^{Blk}$ mice. These data suggest that bone marrow cells contribute to recovering insulin expression in the islets of diabetic mice.

**Fig. 1 The role of bone marrow in diabetic mice. a** Schematic of the study design. Gray mice indicate non-DM and orange mice indicate DM mice. The syringe shows STZ injection while the thunder sign indicates irradiation. Non-DM→DM$^{Ins}$ group ($n = 5$ independent animals), mice that received bone marrow transplantation from non-DM mice after diabetes (DM) had been induced by STZ and received treatment with an insulin pellet (Ins); Non-DM→DM$^{Blk}$ group ($n = 6$ independent animals), STZ-induced DM mice that were treated with a blank pellet (Blk) and then underwent bone marrow transplantation from non-DM mice; DM→DM$^{Ins}$ group ($n$: started with $n = 5$ independent animals, but ended with $n = 2$ at 8 weeks), STZ-induced DM mice that were normoglycemic with insulin prior to the bone marrow transplantation from DM mice; DM→DM$^{Blk}$ group ($n$: started with $n = 5$ independent animals, but ended with $n = 3$ at 8 weeks), STZ-induced DM mice with blank pellets that were reconstituted with bone marrow from DM mice. **b** Serum glucose concentrations in the four groups over the course of 8 weeks. Data are indicated as means ± SE. **c** Blood glucose at 8 weeks in the non-DM→DM$^{Ins}$ and non-DM→DM$^{Blk}$ groups. $**P < 0.01$. Immunohistochemical analysis of insulin (**d**) and glucagon (**e**) in the pancreas. Islets are surrounded by dotted lines. Insulin-positive signals (brown) and glucagon-positive signals (brown) in each islet were quantified in each group. Scale bars = 20 μm, $**P < 0.01$.

**Abnormal BMDCs in the pancreas and thymus.** Based on our previous studies showing that abnormal BMDCs were the cause of the development of diabetic complications[12,14,16–18], we also assume that similar to the case of diabetic complications abnormal BMDCs could also migrate into the pancreas and fuse with resident cells to impair insulin production in the islets. To test this hypothesis, bone marrow cells of GFP-Tg mice were transplanted into nondiabetic mice, and then diabetes was induced by STZ (Fig. 2a). Some cells expressed either GFP (green), a marker of BMDCs, or Vcam-1 (red), a marker for abnormal BMDCs[12], in the islets (Fig. 2b), and double-positive cells for both markers were more frequently observed in diabetic mice than in nondiabetic mice. We next created a mouse model in which the total bone marrow of Ayu1 Cre mice was transplanted into lsl-tdTomato mice (Fig. 2d). In this model, the cell fusion between BMDCs from the donor mice and resident cells in the recipient mice theoretically creates red fluorescence. There were a higher number of cells expressing positive signals for tdTomato (red) in the pancreas of diabetic islets (Fig. 2e) than in that of nondiabetic islets, indicating that the fusion of resident islet cells with BMDCs occurred more frequently in diabetic islets.

To identify the specific types of resident cells fusing with BMDCs in the islets, immunohistochemical analysis of NG2, a marker of pericytes, and von Willebrand factor (VWF), a marker of endothelial cells, was performed. Green signals for NG2 (Fig. 2g) or VWF (Fig. 2h) turned yellow in some cells in the islets of diabetic mice, indicating the overlap between green GFP and red tdTomato; therefore, these BMDCs could fuse with pericytes and endothelial cells in the islets of diabetic mice. Given these results, the fusion of abnormal BMDCs with pericytes or endothelial cells could disturb vascular niche formation, which could be a potential mechanism by which diabetes impairs tissue regeneration in the islets in diabetes.

Next, we examined the thymus based on a previous report showing that bone marrow disorders were closely associated with thymic dysfunction[19]. Interestingly, we found that the thymus was atrophic in diabetic mice compared to nondiabetic mice (Supplementary Fig. 1a–c). We then examined the involvement of BMDCs in the thymus of nondiabetic and diabetic mice. Similar to the pancreatic islets, several GFP- and/or Vcam-1-positive cells were observed in the thymus of nondiabetic and diabetic mice, and the cells expressing both signals were more frequently observed in diabetic mice than in nondiabetic mice (Fig. 2c). Likewise, the expression of tdTomato was very faint in nondiabetic mice while several cells (arrowheads) showed strong red signals in diabetic mice (Fig. 2f), indicating that BMDCs were fused with resident cells in the thymus in diabetic mice (white arrows indicate nonspecific signals).

Finally, we performed FACS analysis of total thymic cells from control nondiabetic mice and STZ-induced diabetic mice. We found that there was no difference in the percentage of total T cell number in the total thymic cells between control nondiabetic mice and STZ-induced diabetic mice (10.8 ± 1.8% vs. 9.9 ± 0.9%, respectively). Dot plot diagrams showed that the fractions of CD4+ and CD8+ cells in total CD3+ cells in control nondiabetic mice are also similar to those

in STZ-induced diabetic mice (Supplementary Fig. 2a, b). Likewise, immunohistochemistry showed that the number of CD8a in the thymus of nondiabetic mice was not different from those in diabetic mice (Supplementary Fig. 2c).

**Replacing diabetic BMDCs with physiological BMDCs allows the islet and thymus to recover from diabetic injury.** We next investigated the mechanism as to how healthy bone marrow contributed to the remission of diabetes using GFP-tg mice as a donor to monitor BMDCs in the recipient (Fig. 3a). After induction of diabetes with STZ, diabetic mice were divided into two groups: one group received blank pellets as uncontrolled diabetes, while the other group was treated with insulin pellets. These mice underwent BMT from nondiabetic GFP-tg mice (Fig. 3a). Assumingly, the transplanted BMDCs would be exposed to high glucose in uncontrolled diabetic mice (GFP$^{non-DM}$→DM$^{Blk}$), so that those cells would turn to be deleterious. In contrast, the transplanted BMDCs would exhibit healthy physiological activity when those cells were under glucose control in insulin-treated diabetic mice (GFP$^{non-DM}$→DM$^{Ins}$). Immunofluorescent analysis for Vcam-1 was performed to detect abnormal BMDCs. As shown in Fig. 3b, the islets in GFP$^{non-DM}$ → WT non-DM mice showed no cells expressing Vcam-1 (red). In contrast, both uncontrolled diabetic mice in the GFP$^{non-DM}$ → WT DM group and GFP$^{non-DM}$→DM$^{Blk}$ group showed some yellow cells in islets, suggesting that uncontrolled diabetes was associated with the infiltration of abnormal BMDCs in pancreatic islets. However, the combination therapy of insulin and BMT from nondiabetic mice eliminated the invasion of these abnormal BMDCs in GFP$^{non-DM}$→DM$^{Ins}$ mice (Fig. 3b).

After the thymus was confirmed to be atrophic in diabetic mice compared to nondiabetic mice (Supplementary Fig. 1), we then examined the effect of the combination therapy on the thymic mass (Fig. 3c). Stereomicroscopy revealed green fluorescence in the whole mass of the thymus in control GFP$^{non-DM}$ → WT non-DM mice while diabetic mice in the GFP$^{non-DM}$ → WT DM group and GFP$^{non-DM}$→DM$^{Blk}$ group had a shrunken thymus with reduced GFP expression (Fig. 3c). However, the combination of insulin therapy with BMT from nondiabetic mice was able to restore the size of the thymic mass (GFP$^{non-DM}$→DM$^{Ins}$) (Fig. 3c). Likewise, histological analysis of the thymus revealed that compared with control GFP$^{non-DM}$ → WT non-DM mice, a few GFP-positive cells were detected in diabetic mice with uncontrolled glucose levels in the GFP$^{non-DM}$ → WT DM group and GFP$^{non-DM}$→DM$^{Blk}$ group. These data indicate that the progeny of abnormal BM cells, including those of the T-cell lineage[12], might fail to migrate into the thymus, perhaps due to a loss of physiological function. In contrast, there were many GFP-positive cells in diabetic mice treated with combination therapy (Fig. 3d).

**Blocking HDACs together with insulin induces complete remission in diabetes.** Based on our previous study showing that Vcam-1$^+$ST-HSCs, which are causally involved in the development

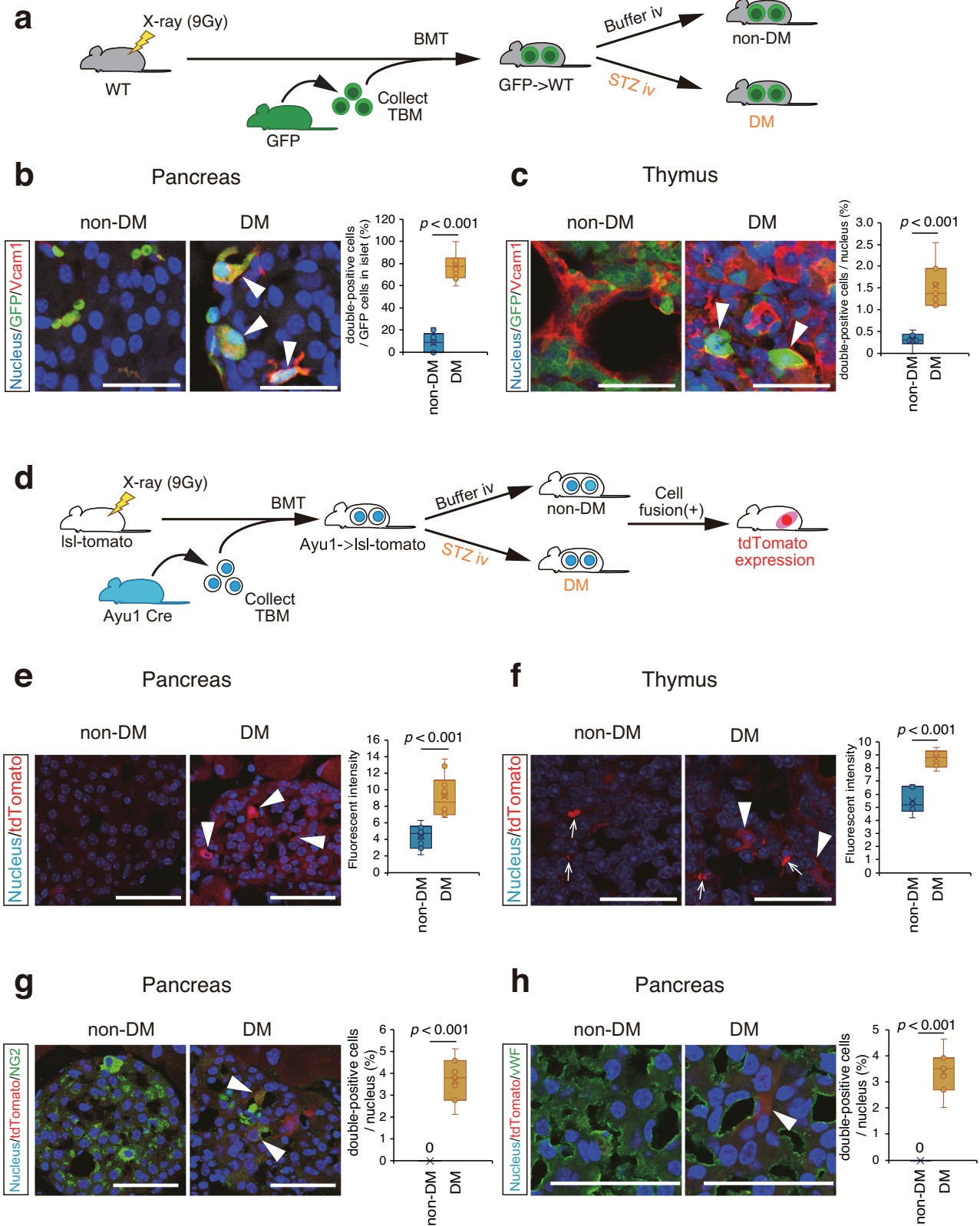

of diabetic complications, belonged to Lin(-)Sca-1(+)c-kit(+) cells (LSK cells) in the BM[12], LSK cells were then isolated to examine the expression patterns of histone deacetylases (HDACs). The expression of some types of HDACs was markedly upregulated, including type-1, -3, -4, and -8 and sirt-1, -6, and -7, while some were downregulated, such as type-2, -9, and -11, suggesting that

diabetes caused epigenetic changes in LSK cells of the BM (Supplementary Fig. 3).

We then hypothesized that epigenetic modifications could eliminate pathological changes in BMDCs and cure diabetes. As shown in Fig. 4a, STZ-induced diabetic mice were transiently treated with insulin pellets only for an initial couple of weeks, while

**Fig. 2 Pathological roles of BMDCs in the pancreas and thymus. a** The schema of the study design. Gray mice indicate WT mice and green mice show GFP transgenic mice. Non-DM mice ($n = 4$ independent animals), wild-type mice that were reconstituted with bone marrow from GFP transgenic mice; DM mice ($n = 4$ independent animals), diabetes was induced by STZ in mice that had been reconstituted with bone marrow from GFP transgenic mice. **b** Vcam1 expression (red) in GFP-positive cells (green) of the pancreas in non-DM and DM mice. Yellow signals indicate double positivity with red (Vcam1) and green (GFP). Quantification of the percentage of double-positive cells among GFP-positive cells is shown in the right panel. **c** Immunofluorescence analysis shows GFP (green) and Vcam1 (red) expression in the thymus. The number of double-positive cells (yellow) was higher in DM mice than in non-DM mice. **d** The schema of the study design. White mice indicate WT mice and blue mice show Ayu1-Cre transgenic mice. After radiation, lsl-tomato mice received bone marrow from Ayu1-Cre mice before they were intravenously administered either citrate buffer or streptozotocin to create non-DM ($n = 4$ independent animals) and DM mice ($n = 4$ independent animals). In this design, the fusion of Ayu1-Cre-positive bone marrow-derived cells with resident cells of lsl-tomato mice theoretically results in red color expression. **e, f** Immunofluorescence of tdTomato as red signals (arrows) is often seen in the pancreas (**e**) and thymus (**f**) of DM mice but not in those of non-DM mice. **g, h** Immunofluorescence analysis of NG2 (green in (**g**)), a marker of pericytes, and VWF (green in (**h**)), an endothelial marker, is shown. Arrowheads show yellow signals, indicating double positivity for red (tdTomato) and green (NG2 or VWF). Scale bars = 50 μm.

those mice were treated with givinostat, a nonspecific HDAC inhibitor, for 8 weeks (Fig. 4a, b). Givinostat was administered either orally or by venous injection because it was uncertain which method could exert therapeutic efficacy at that moment. Importantly, the therapeutic efficacy continued to be observed for an additional 4 weeks without any treatment after the end of givinostat administration, with the hope that normal glucose levels would be maintained without any therapy if diabetes could be put into remission (Fig. 4a, b). Givinostat alone, by either oral administration (DM$^{Blk+Giv\ p.o.}$ shown by the dark-blue line) or by venous injection (DM$^{Blk+Giv\ i.v.}$ shown by the green line), did not lower glucose levels in diabetic mice (Fig. 4b). Diabetic mice treated only with insulin pellets exhibited normoglycemia only during the first 2–3 weeks while insulin pellets were present, but after insulin pellets faded out, the blood glucose concentrations gradually increased later in diabetic mice (DM$^{Ins}$ shown by the orange line). However, glucose control was maintained in diabetic mice that were administered givinostat for 8 weeks with an initial insulin pellet implantation (DM$^{Ins+Giv\ i.v.}$ shown by the red line and DM$^{Blk+Giv\ p.o.}$ shown by the purple line). Importantly, these two groups remained normoglycemic even after the insulin pellet had faded. More importantly, these mice also showed sustained normoglycemia for an additional 4 weeks even after the withdrawal of givinostat treatment (Fig. 4b, c). Mean glucose concentrations between 9 and 12 weeks, during which there was no therapy, were statistically lower in diabetic mice with the combination therapy than in untreated diabetic mice (Fig. 4c). These data support our assumption that diabetic remission could be achieved by a transient HDAC inhibition only when initial glucose control is well controlled. Of note, several diabetic mice receiving blank pellet and givinostat orally or by injection were dead at 8 weeks (blue and green lines), while all animals survived in the other groups, suggesting that monotherapy with givinostat was deleterious in diabetic mice.

To examine the therapeutic efficacy of the combination therapy, the expression levels of endogenous insulin (Fig. 4d, e) and glucagon (Fig. 4d, f) were immunohistochemically examined at the end of the study. Compared to nondiabetic mice, untreated diabetic mice exhibited markedly reduced insulin levels in islets. Monotherapy with insulin pellets (DM$^{Ins}$) or givinostat (DM$^{Blk+Giv\ i.v.}$, DM$^{Blk+Giv\ p.o.}$) did not change the signal intensities of these two factors, while the combination therapy increased insulin signals (DM$^{Ins+Giv\ i.v.}$, DM$^{Ins+Giv\ p.o.}$). In contrast, glucagon signals in the islets were significantly higher in untreated diabetic mice than in nondiabetic wild-type mice (Fig. 4e), while the combination therapy succeeded in lowering glucagon signals (DM$^{Ins+Giv\ i.v.}$, DM$^{Ins+Giv\ p.o.}$) (Fig. 4e). Likewise, serum insulin concentrations in diabetic mice with the combination therapy also returned to the levels of nondiabetic mice (Supplementary Fig. 4).

In terms of the thymus (Fig. 4g, h), the organ appeared to be atrophic in untreated diabetic mice (DM) compared to nondiabetic

mice (non-DM). Similar to the changes in insulin and glucagon expression, monotherapy with insulin pellets (DM$^{Ins}$) or givinostat (DM$^{Blk+Giv\ i.v.}$, DM$^{Blk+Giv\ p.o.}$) failed to show any changes, but the thymus size recovered in response to the combination therapy in the treated diabetic mice (DM$^{Ins+Giv\ i.v.}$, DM$^{Ins+Giv\ p.o.}$).

**TNF-α is expressed in abnormal BMDCs to mediate diabetic injury in the BM and thymus.** A key mechanism by which diabetes causes pancreatic and thymic injuries would be accounted for by the deleterious ability of abnormal BMDCs to express the inflammatory cytokine TNF-α[18] and several chemokines (Supplementary Fig. 5). The untreated diabetic mice markedly expressed TNF-α in BM cells and the thymus but not in the pancreatic islets compared with nondiabetic mice (Supplementary Fig. 6). A potential reason for the lack of TNF-α expression in pancreatic islets might be that the β cells had already been largely destroyed by STZ, leaving no β cells to express the cytokine. Single therapy with givinostat (DM$^{Blk+Giv\ i.v.}$, DM$^{Blk+Giv\ p.o.}$) or insulin pellets (DM$^{Ins}$) was not sufficient in blocking TNF-α expression. In turn, the combination therapy succeeded in reducing TNF-α expression in these organs (DM$^{Ins+Giv\ i.v.}$, DM$^{Ins+Giv\ p.o.}$) (Supplementary Fig. 6).

**The thymus is the key to the remission of diabetes.** Our data indicate that the thymus may be involved in the pathogenesis of diabetes and could play a key role in the treatment of diabetes. To determine the role of the thymus, we surgically removed this organ in diabetic mice treated with combination therapy (Fig. 5a). As shown in Fig. 5b, thymectomy significantly eliminated the antidiabetic effects of this treatment. These mice were sacrificed at 8 weeks, and we confirmed that the thymus was successfully removed in these mice (Fig. 5c).

## Discussion

Our previous study demonstrated that a cause of diabetic complications was Vcam1$^+$ST-HSCs in the bone marrow, which pathologically express proinsulin and TNF-α and fuse with resident cells in several organs, including neurons, the liver, adipose tissue, and the thymus, and impair their physiological functions[13]. Here, we hypothesized that the same mechanism could be driven in the pancreas to disturb β cells and impair insulin expression. Importantly, the fact that those abnormal BMDCs could not be eliminated by controlling serum glucose indicates that those cells could account for the diabetic refractoriness to current therapy. Thus, these cells could be a novel therapeutic target to eliminate diabetes.

To test our hypothesis, we first attempted to replace abnormal BMDCs with healthy BMDCs in diabetic mice. In addition, we also thought that the blood glucose concentration should be controlled by insulin to block the new production of abnormal

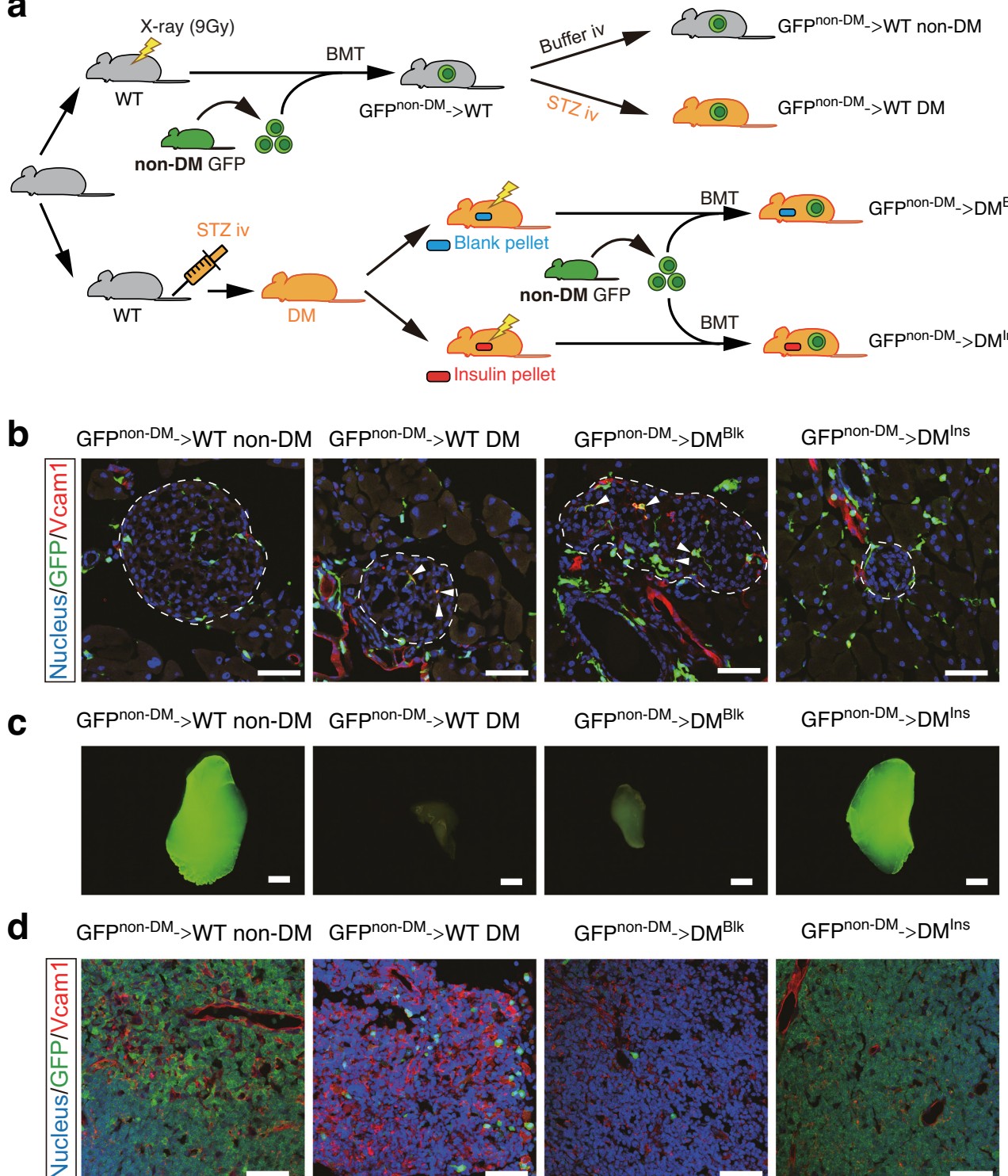

**Fig. 3 The combination of insulin with BMT blocks the infiltration of abnormal BMDCs in the pancreas and thymus of diabetic mice. a** The schema of the study design. Gray mice indicate non-DM and orange mice indicate DM mice. The syringe shows STZ injection while the thunder sign indicates irradiation. GFP[non-DM] → WT non-DM ($n = 4$ independent animals) and GFP[non-DM] → WT DM groups ($n = 4$ independent animals), bone marrow of nondiabetic (non-DM) GFP transgenic mice was transplanted into wild-type (WT) mice before non-DM or DM mice were induced with citrate buffer or streptozotocin, respectively. In the GFP[non-DM]→DMBlk ($n = 4$ independent animals) and GFP[non-DM]→DMIns groups ($n = 4$ independent animals), wild-type mice were first induced to be diabetic with streptozotocin and were treated with insulin or blank pellets, and then these mice received bone marrow from nondiabetic GFP transgenic mice. **b** Immunofluorescence of GFP (green) and Vcam1 (red). Yellow (arrowheads) shows the overlap of GFP with Vcam1. Dotted lines indicate islets of the pancreas. Scale bars = 50 μm. **c** Macroscopic GFP expression in the thymus. Scale bars = 10 μm. **d** Immunofluorescences of GFP (green) and Vcam1 (red) in the thymus. Scale bars = 50 μm.

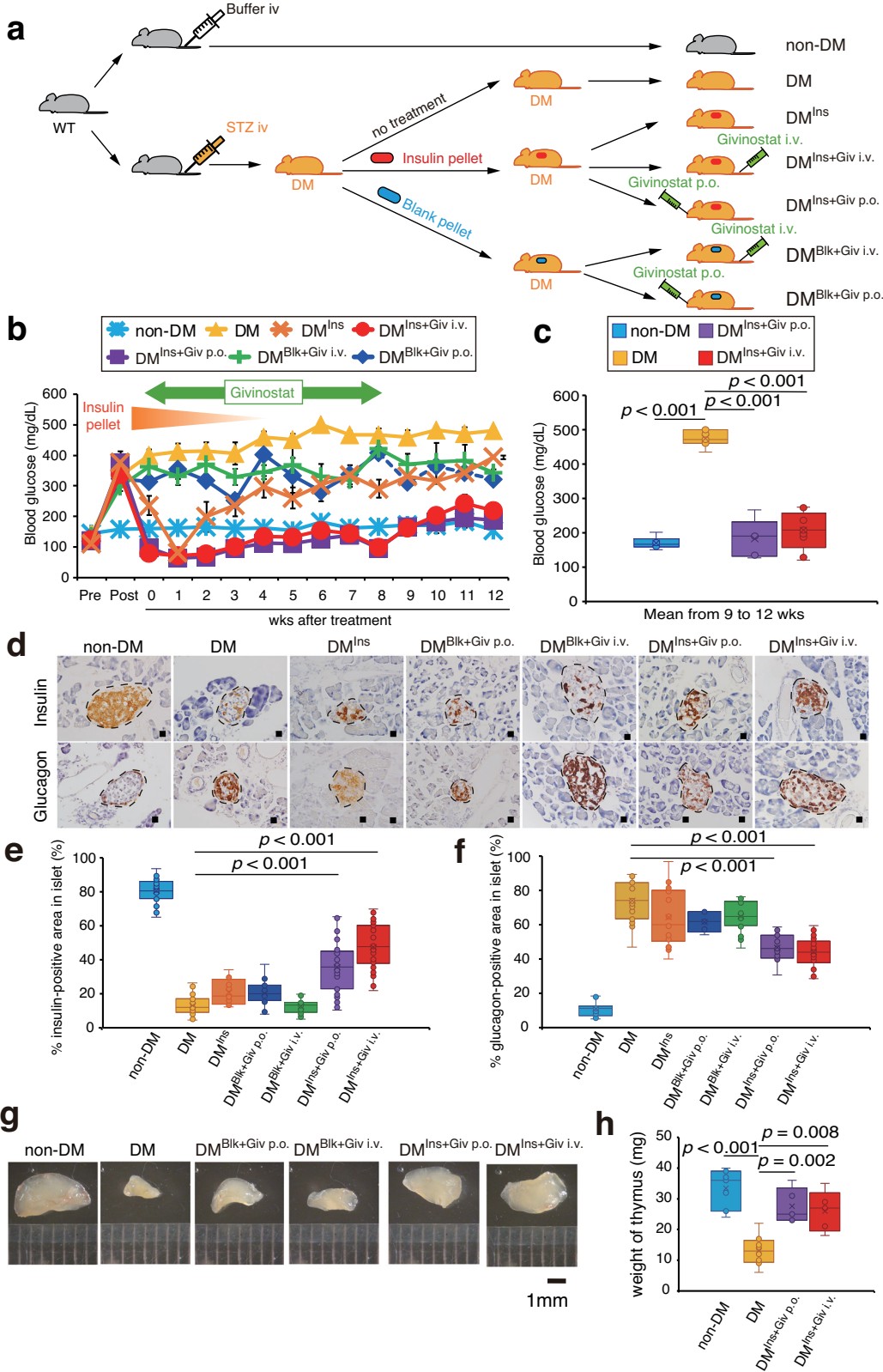

BMDCs, as those abnormal BMDCs continued to be produced under high glucose concentrations[12]. As we expected, BMT with insulin pellets succeeded in maintaining normoglycemia, which importantly was sustained even after the insulin pellets had faded away. We confirmed that this sustained therapeutic effect would account for restoring endogenous insulin production in the pancreas. This experiment clearly demonstrated that a

cause of diabetes was abnormal BMDCs, which could be a therapeutic target.

As an alternative to BMT, a pan-HDAC inhibitor, givinostat, was administered based on our data showing that abnormal BMDCs exhibited epigenetic changes with aberrant HDAC expression. The pharmaceutical inhibition of HDAC activity exhibited antidiabetic effects similar to those of BMT. In fact,

**Fig. 4 Sustained normoglycemia is achieved without any treatments in diabetic mice. a** The schema of the study design. Gray mice indicate non-DM and orange mice indicate DM mice. White syringe shows control buffer injection whereas orange and green syringes indicate STZ and givinostat injections, respectively. After the induction of diabetes using STZ, diabetic mice were treated with no treatment ($n = 4$ independent animals), blank ($n = 8$ independent animals) or insulin pellets ($n = 13$ independent animals). At the same time, some were also treated with givinostat orally or by injection. **b** Time course of blood glucose concentrations for 12 weeks. Data are indicated as means ± SE. **c** Mean blood glucose concentrations between 9 and 12 weeks, which were after the end of combination therapy ($n = 5–9$ per group. Independent measurement) *$P < 0.05$. **d** Immunohistochemistry for insulin and glucagon in the islets at 12 weeks; brown color indicates insulin (upper panels) or glucagon signals (lower panels). Black dotted lines surround the islets of the pancreas. Scale bars = 20 μm. **e, f** Quantification of the % insulin-positive area (**e**) and glucagon (**f**) in an islet, is shown. ($n = 2–5$ per group. Independent measurement). **g, h** Macroscopic thymic masses (**g**) and the quantification of thymic mass weight (**h**) ($n = 5–8$ per group. Independent measurement) are shown. Scale bars = 1 mm. **$P < 0.01$.

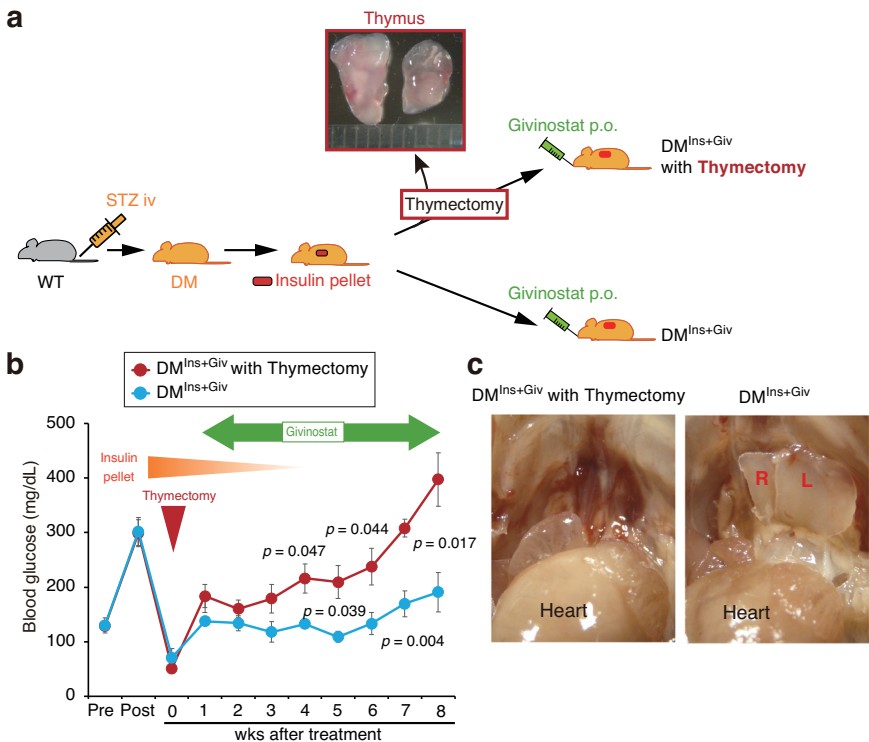

**Fig. 5 The thymus is a key mediator in the progression of diabetes. a** The schema of the study design. Gray mice indicate non-DM and orange mice indicate DM mice. A syringe with orange or green color indicates STZ or givinostat injections, respectively. Eight diabetic mice were treated with an insulin pellet and then divided into two groups: one group underwent thymectomy ($n = 4$ independent animals), and the other group underwent sham surgery ($n = 4$ independent animals). Both groups were treated with givinostat for 8 weeks after thymectomy. **b** Time course of the blood glucose concentration for the entire period of this study. Data are indicated as means ± SE. **c** Representative pictures of the thymus in the two groups at sacrifice. R and L indicate the right and left lobes of the thymus, respectively.

givinostat maintained normoglycemia after the end of insulin therapy in diabetic mice, and more importantly, the effects of HDAC inhibition were sustained for at least 4 weeks after the cessation of both givinostat and insulin. These data suggest that the epigenetic changes in abnormal BMDCs would be a key mechanism for diabetic refractoriness and that HDAC inhibition could eliminate diabetic epigenetic changes, leading to complete remission for diabetes.

Here, the present study demonstrated that there were several mechanisms by which diabetes interfered with the regeneration of islets. First, high glucose-induced aberrant HDAC activities were associated with the upregulation of several pathological factors, including proinsulin and inflammatory cytokines and chemokines, in abnormal ST-HSCs in diabetes. Since those cells retain the ability to fuse with resident cells[13], the fused cells are likely exposed to inflammatory cytokines and chemokines that disturb their physiological functions. In particular, the fusion of those abnormal BMDCs with endothelial cells or pericytes would be

critical, as endothelial cells and pericytes are key components of vascular niche formation for tissue regeneration[20]. Such a pathological process might be a key mechanism for interfering with islet regeneration under diabetic conditions.

Another interesting finding was thymic atrophy in diabetic mice, and insulin therapy failed to alleviate the pathological changes. The potential mechanism for thymic atrophy would be accounted for by abnormal BMDCs homing to the thymus to induce inflammation. We have long wondered why aberrant BMDCs, despite the non-physiological expression of proinsulin and TNF-α, are not considered "a foreign substance" to be eliminated by the immune system in the context of diabetes. However, the present study might provide a clue for understanding that the atrophied thymus loses its physiological function and allows aberrant BMDCs to escape from immune recognition. Importantly, the atrophic thymus is likely a key determinant of the therapeutic effect. Since surgical removal of the thymus was shown to completely eliminate the protective

effects of these therapies, the thymus may be indispensable to diabetic remission. This finding is consistent with recent studies showing that the thymus is not simply an immune system that educates T cells but also contributes to the tissue regeneration process. In particular, the subset of regulatory T cells contributes to tissue regeneration[21].

Previously, our group identified a subset of ST-HSCs that aberrantly expressed Vcam-1 and TNF-α and home to various organs, including neurons, the liver, adipose tissue, and the thymus, where those cells fuse with residential cells and may play a detrimental role in driving diabetic complications[12]. While cell fusion between BMDCs and other cells under nondiabetic conditions is usually a physiological process[22,23], it is likely detrimental when the BMDCs are impaired and can induce irreversible damage under diabetic conditions.

In contrast to BMT, which is an unlikely practical therapy for each diabetic patient in clinical medicine, an HDAC inhibitor with insulin could be an alternative and practically utilized. HDAC inhibitors are not currently used to treat diabetic patients but have been used in clinical practice to treat several diseases, including hematological malignancies and multiple sclerosis[24], suggesting that they are safe.

In our hypothesis, high glucose is the key factor driving the progression of diabetes, suggesting that this mechanism could be shared by both type 1 and type 2 diabetes and that the therapy can be effective for both types. In other words, we need to be aware that the present therapy inhibits only the driving machinery but does not block the initiation process in both types of diabetes.

In conclusion, the present study demonstrated that complete remission in diabetes can be induced by the combination of HDAC inhibition and blood glucose control. Eliminating aberrant BMDCs could restore the regenerative ability of β cells for insulin secretion (Fig. 6).

## Methods

**Animal models**. All experimental protocols in this study were approved by the Animal Care Committees of the Shiga University of Medical Science to comply with all relevant ethical regulations for animal testing. C57BL/6J mice (wild type, Japan SLC, Inc., Shizuoka, Japan), C57BL/6-Tg (UBC-GFP) 30Scha/J mice (GFP-Tg mice, The Jackson Laboratory, Bar Harbor, ME, USA), B6.Cg-Gt(ROSA) 26Sor^tm9(CAG-tdTomato)Hze/J (Lsl-tdTomato mice, The Jackson Laboratory) and Ayu1 promoter-driven Cre recombinase-expressing (Ayu1-Cre) mice[25] were used.

In the first experiment, diabetic mice were reconstituted with the BM of wild-type mice to examine whether the BM of wild-type mice has an antidiabetic effect. Diabetes was induced by the injection of STZ (150 mg/kg) into the tail vein, and control mice were administered citrate buffer (pH 4.5)[12,13]. The induction of diabetes was confirmed by a serum glucose concentration >250 mg/dl one week after STZ injection. Diabetic mice that had been diabetic for 12 weeks or age-matched wild-type mice were used as donors for BMT. An insulin pellet or a blank pellet (LinBit, LinShin Canada, Inc., Canada) was subcutaneously inserted into some diabetic mice under isoflurane inhalation anesthesia. Basically, an insulin pellet lasts for only a couple of weeks, so after 4 weeks, blood glucose is controlled by endogenous insulin.

The second experiment included two sets of mice: recipient C57BL/6J mice that received BM from donor GFP-Tg mice and floxed-tomato mice reconstituted with BM cells from Ayu1-Cre mice. If cell fusion occurred between cells in floxed-tomato mice and BMDCs from Ayu1-Cre mice, the fusion cells would develop red fluorescence because Cre recombinase in the BM cells of Ayu1-Cre mice reacts with the lox-stop-lox codon in the floxed-tomato mice.

The third experiment included two types of recipients (Fig. 3a). One group included mice that were reconstituted with BM cells from wild-type GFP transgenic mice and injected with buffer or STZ to create control or diabetic mice, respectively. In the other group, diabetes was initially induced with STZ, and then the mice were treated with a blank or insulin pellet prior to receiving BM cells from nondiabetic GFP-Tg mice.

In the next experiment, 1 week after STZ-induced diabetic mice were treated with either a blank or an insulin pellet, 2.5 mg/kg givinostat (Cayman Chemical, Michigan, USA), a pan-HDAC inhibitor, was administered orally or by injection via the tail vein every 3 days for 8 weeks[26].

In the final experiment, thymectomy was performed in treated diabetic mice[27] to examine the involvement of the thymus in the pathophysiology of diabetes. Briefly, mice were placed in a supine position and fixed to the stereomicroscope. After a

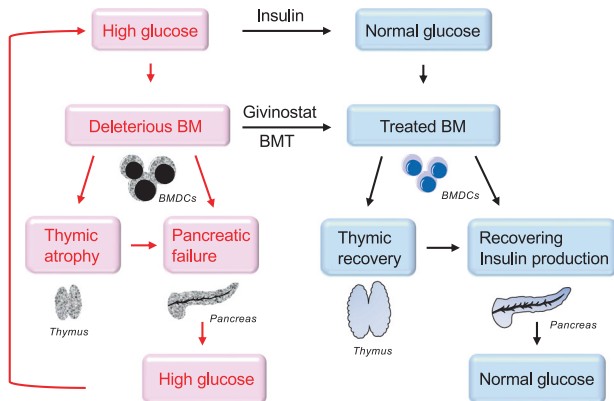

**Fig. 6 The conceptual schema shows the mechanism for the progression of diabetes and how combination therapy could block the deleterious pathways.** High glucose initiates BM dysfunction, resulting in both pancreatic and thymic dysfunctions. Pancreatic dysfunctions sustain hyperglycemia while thymic dysfunction could lead to pancreatic dysfunction. In order to block the vicious cycle, both glucose control and HDAC inhibition are required. Blood controls could eliminate the future production of abnormal BM cells whereas an HDAC inhibition would remove the diabetic memory from existing deleterious BM cells.

median incision was made in the anterior neck and the connective tissue around the thyroid gland was detached, the thymus was carefully and surgically removed[27].

**Bone marrow transplantation**. The recipient mice were exposed to 9 Gy of radiation by an MBR-1520R X-ray irradiator (Hitachi Medical Corporation, Tokyo, Japan) and then underwent BMT as described previously[12]. We confirmed that BMT reconstituted 95% of peripheral white blood cells of donor origin[12].

**Immunofluorescence analysis**. After exsanguination, the mice were perfused with 4% paraformaldehyde in 0.1 M phosphate buffer, and the pancreas, thymus and bone were collected. Rabbit anti-insulin (Cell Signaling Technologies, Danvers, MA USA), rabbit anti-glucagon (Cell Signaling Technologies) and rabbit anti-TNF-α (Abcam, Cambridge, UK) were used as primary antibodies, followed by Imm-PRESS reagent anti-rabbit IgG as the secondary antibody. The color was developed using an ImmPACT DAB kit (Vector Laboratories, Burlingame, CA, USA). For immunofluorescence analysis, the sections were incubated with anti-Vcam-1 (Cell Signaling Technologies) or rabbit anti-NG2 (Proteintech, USA) and sheep anti-von Willebrand Factor (Abcam) primary antibodies at 4 °C overnight, followed by Alexa555 anti-rabbit IgG as a secondary antibody (Thermo Fisher Scientific Inc. Waltham, MA, USA). To examine the localization of CD8a T cells in the thymus, frozen sections of fixed thymus were stained with rat anti-CD8a (Clone 53-6.7, Biolegend) as primary antibody and Alexa488 anti-rat IgG (Thermo Fisher Scientific) as secondary antibody at 4 °C overnight. These fluorescent immunostained sections were mounted with a DAPI-containing medium (Vector Laboratories) and observed under a fluorescence microscope.

**Evaluation of the thymus**. After being perfused with fixative, the thymus was gently removed using a stereomicroscope. The organ was immersed in 15% sucrose in 0.1 M phosphate buffer and then weighed. Images were captured and examined by a fluorescence stereomicroscope.

**FACS analysis of thymus**. Thymus was isolated from nondiabetic and diabetic mice perfused with PBS(−) after anesthesia to prepare the suspensions of the thymocyte. Thymocyte suspensions were stained with LIVE/DEAD violet dead cell stain kit (Thermo Fisher Scientific) for 30 min to remove dead cells, washed with PBS(−), and reacted with anti-CD16/32 antibody (Clone 93, Biolegend) to block Fc receptor. Cell suspensions were then stained with PECy7 CD3 (Clone 17A2, Biolegend), FITC CD4 (Clone GK1.5, Biolegend), and APC CD8a (Clone 53-6.7, Biolegend) antibodies for 30 min. The suspensions of stained cells were analyzed using FACS Aria Fusion (BD bioscience). A portion of unstained suspended cells was stained with an isotype control of each fluorescent antibody to determine gating. Acquired data were analyzed using FACS DIVA software (BD Bioscience).

**Microarray analysis**. Monocytes were isolated from total BM cells by Ficoll-Paque-PLUS (Cytiva, Tokyo, Japan) before dead cells were removed with a LIVE/DEAD Fixable Blue DEAD cell stain kit (Thermo Fisher Scientific Inc.). The samples were stained with a Biotin Mouse lineage panel (BD Biosciences, San Jose, CA, USA), followed by PECy7-conjugated streptavidin, APC-conjugated c-kit, and

APCCy7-conjugated Sca-1 (all from BD Biosciences), and then LSK cells were isolated by a FACS Fusion (BD Biosciences). An RNeasy Plus Micro kit (Qiagen) was used to extract RNA from LSK cells, and then the samples were sent to Takara Bio Inc. (Shiga, Japan), where the RNA was amplified and microarray analysis was performed.

**Statistics and reproducibility**. Data were evaluated using $t$-tests to analyze differences between two groups or using one-way ANOVA to analyze differences among more than three groups. The number of mice for each experiment is provided in the result and in the figure legends. Data are expressed as the mean ± SE. $P < 0.05$ was considered significant.

**Reporting summary**. Further information on research design is available in the Nature Portfolio Reporting Summary linked to this article.

## Data availability
Raw source data for all figures and supplementary figures can be found in Supplementary Data 1. The accession code for microarray data is GSE224690 in Gene Expression Omnibus (GEO).

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

## Acknowledgements
We thank Mrs. Reiko Yamamura and Miss Machiko Yamamoto for administrative assistance and Takefumi Yamamoto and Yasuhiro Mori in the central research laboratory for technical assistance.

## Author contributions
H.K. had the original idea, designed the experiments and discussed all issues as a leader for this study. M.K. performed most experiments and created the figures. J.O., Y.N., N.O., K.F. and I.M. performed some experiments, wrote figures and discussed most issues for this study. T.N. designed some experiments, discussed several issues and wrote the entire manuscript. All authors read and approved the final manuscript.

## Competing interests
The authors declare no competing interests.
