## [Peer Review File · Communications Biology]

Reviewers' comments:

Reviewer #1 (Remarks to the Author):

Reviewer's report

Hideto Kojima et al, investigated that treating Vcam-1+ST-HSCs with a combination of insulin and givinostat, an HDAC inhibitor, would cure diabetes. Authors reported that diabetes results in the fusion of Vcam-1+ST-HSCs with resident cells in pancreatic islets and the thymus, a pathological process that was blocked by the combination of insulin and givinostat. Thymic mass was decreased in diabetic mice and restored by the combination therapy. Authors therefore concluded that diabetes might be a bone marrow disease with disturbed thymus and targeting abnormal bone marrow stem cells could a solution to diabetes. Overall, the study sound interesting, and the results will add to the pool of knowledge in the field. However, there are several concerns authors need to address:

Major comments

1. Abstract: Authors need to re-write the abstract and clearly state the aim or hypothesis of the study. The methodology should be described briefly in the abstract.
2. Introduction: Authors should rewrite the introduction and update their references on the pathophysiology and management of diabetes and clearly state the justification for this study.
3. Authors should rewrite the whole manuscript for scientific publication
4. How did authors determine the dose of STZ? How was the dose of insulin and givinostat determined? Why did authors give givinostat every 3 days for 8 weeks? Authors need to state the rationale for choosing the dose, route and duration of administration for reproducibility.
5. Did authors determine the wet or dry weight of thymus?
6. Why did authors represent their data with SE rather than SD?
7. Authors should include in their discussion if any, the difference between givinostat and endogenous HDAC inhibitors like short chain fatty acids (butyrate or propionate)? Which one seems superior?
8. Authors stated in the discussion that "the evidence in this study shows that islets that had failed to produce insulin regained insulin expression in response to the combination therapy, which indicates that β -cells retain the ability to regenerate islets" Was the effect of the combination therapy restorative or ameliorative on the islets? This should be made clear in your discussion to guide the readers right.
9. Was there any side effect (s) noticed with BMT?
10. Authors should correct all the punctuation, typographical and grammatical errors in the manuscript.
11. The manuscript will also benefit from the service of a professional language editor.

Reviewer #2 (Remarks to the Author):

The authors provided a detailed study on how epigenetic modification in bone marrow stem cells is involved in diabetes progression and the regulation of this modification not only stop the progression but also can cure diabetes. The manuscript is well designed and provided the clear insights about the role of BM in diabetes pathogenesis by showing the involvement of normal BM in regulating the normoglycemia phase in glucose-controlled environment. The study also highlights the mechanistic approach of deteriorating insulin secretion from islets by the movement of aberrant BMDCs which state a conclusive remark about the negative impact of abnormal BMDCs migration in β cell regeneration in diabetic condition. The authors proposed removal of abnormal BMDCs as a crucial approach to ameliorate the pathogenesis of diabetes. The authors hypothesized on the ground of differential expression of HDACs in pathogenic BM cells that epigenetic modulation could provide the alternative of impractical BMT to each individual and experimentally provided significant effect of nonspecific HDAC inhibitor along with early glucose control maintained normoglycemia for a period of 8 weeks and carried over to an additional 4 weeks even after inhibitor administration, which is interesting. Interestingly the combination study showed significant regeneration of islet forming β

cells. Additionally, the study demonstrated the importance of thymus in regulation of diabetes. The study is well planned and addressed an important aspect which is novel and merits publication.

We request the authors to address following concerns:

- a. How authors decided on using givinostat? Is this educated guess, or they have carried out some sort of screening?
- b. Reviewer requests the author to discuss alternative HDAC inhibitor like entinostat or Panobinostat that are reportedly much milder. Especially, Entinostat has been reported by many authors for its role in metabolic disorders. A discussion in this regard will enrich manuscript.
- c. The reviewer would suggest a RNA seq of BMDC with givinostat (if it is not cost prohibitive). Alternatively, a gene expression analysis on the related signaling pathways will enrich the data.
- d. The authors showed 8 weeks of study upon administration of givinostat followed by a four weeks' observation without any treatment. A chronic therapeutic outcome observation should be appropriate to ensure the long-term effect of givinostat upon limited administration as a trend of gradual increase of blood glucose that can be observed after 12th week of treatment (Fig.4b). The chronic study also will be appropriate concerning the deleterious effect of givinostat and a long-term repetitive treatment with givinostat may rise concern.
- e. The involvement of specific class of HDAC will give more appropriate mechanistic approach and will help in selection of safer and better inhibitor.

Having said this, the reviewer noted the novelty of the approach and brilliant execution of the experiments and recommend publication.

Reviewer #3 (Remarks to the Author):

In their manuscript "HDAC inhibition resets the bone marrow and thymus to cure diabetes", Kojima and colleagues use bone marrow transplantation experiments to investigate the involvement of the latter in type 1 diabetes mellitus (DM1) pathogenesis. They suggest that BM cells drive DM1 pathology and that HDAC inhibition is sufficient to "cure" the disease.

While the authors apparently spent a lot of effort and time on this project, most conclusions in this manuscript are overstated, not supported by experimental data and misleading. This includes the title of the manuscript, which is inappropriate given the experimental design and findings shown.

Additionally, the story line is convoluted and hard to follow. In summary, this study is not suitable for publication in the present form and would require extensive editing/revisions.

Please see some major concerns below.

- As mentioned above, several points in this manuscript are overstated. Headings like "Aberrant BM is the cause of diabetes" or statements suggesting having discovered the "cure" for the disease are not appropriate whatsoever, particularly with respect to the data shown. This also holds true for "The thymus is key to curing diabetes" and so on.
- The authors solely rely on a single animal model that does not recapitulate the true pathogenesis of DM1. Streptozotocin is a toxin directly injuring beta cells, while patients suffering from type 1 diabetes lose beta cells due to autoimmune mediated destruction. If the authors really want to propose having found the cure for a disease, they should use an animal model relevant to the underlying pathology such as NOD mice. This is particularly relevant given the mechanism proposed
- Throughout the study, authors rely on immunohistochemistry analysis to evaluate insulin and glucagon expression in the pancreas, while circulating insulin or c-peptide levels are never shown. This is a weakness of the study
- The study focuses on contributions of the BM to DM1 pathogenesis and the authors use the term "BMDCs" throughout the manuscript. What are "BMDC cells"? how is this population defined? Why did the authors not narrow this down to a defined cell type?
- How do BMDCs or resident immune cell populations including T-cells respond to streptozotocin? It is

unclear whether streptozotocin directly elicits autoimmunity, which would be interesting but irrelevant to DM1 pathogenesis as it is a non-physiological toxin (see previous point)

- The authors conclude that diabetes is a bone marrow derived disease because they observe that blood glucose levels rise in diabetic mice upon cessation of insulin supplementation following transplantation of bone marrow from streptozotocin-exposed animals. This conclusion is a logical fallacy and not supported by the data shown. To actually prove this, the authors would need to transplant bone marrow from diabetic mice to healthy mice (diabetic -- > non-diabetic) and observe that these animals develop diabetes without any additional intervention. These data are crucial and not included in the manuscript. Even if the authors could show this one should still be cautious classifying diabetes as a "bone marrow" disease, which is an overstatement. Moreover, this would not rule out a direct STZ effect (see previous points)
- How do the authors dissect differences between endogenous (beta cell) and exogenous (pellet) insulin production?
- Immunofluorescence in figure 2 does not look convincing and should be statistically quantified
- "yellow"/fused cells are apparently barely detectable in the pancreas of the mice, which questions their relevance in disease pathogenesis
- Overall, it is unclear which relative contributions are attributable to 1.) "cell-fusion", 2.) direct STZ effects on immune cell populations, 3.) thymic dysfunction, 4.) blood glucose and 5.) defined immune cell populations to the phenotype
- Do BMDCs isolated from diabetic mice express an "autoaggressive" gene signature?
- At one point, authors jump from the bone marrow to the thymus, which is confusing to the reader. The thymus is the main tissue for T-cell education, but T-cells are not discussed as relevant players in the first place (which they should). It is also unclear whether "BMDCs" include T-cells (likely yes) and if so, whether depleting T-cells prior to Tx prevents phenotype development
- If the authors suggest that DM1 is a BM-fueled disease, why would it then be surprising that Tx of wildtype BM is required to eliminate "aberrant BM" cells?
- The experimental design in Fig. 4 is unclear. Did the authors inject STZ together with givinostat? If so, this would suggest that children with type 1 diabetes would need givinostat therapy even before the onset of diabetes as STZ-induced diabetes takes days to develop
- The authors found that givinostat monotherapy entailed mortality in WT mice. This is a knock-out criterium for translating these findings into a clinical context
- Although the finding that thymectomy blunts the therapeutic effect of givinostat is interesting, it is unclear how this fits within context of the study hypothesis as the thymus and bone marrow harbor distinct immune cell populations (see previous responses)
- Methods are inappropriately short and do not allow recapitulation of experimental procedures
- Language editing is recommended

Reviewer #1 (Remarks to the Author):

Reviewer's report

Hideto Kojima et al, investigated that treating Vcam-1+ST-HSCs with a combination of insulin and givinostat, an HDAC inhibitor, would cure diabetes. Authors reported that diabetes results in the fusion of Vcam-1+ST-HSCs with resident cells in pancreatic islets and the thymus, a pathological process that was blocked by the combination of insulin and givinostat. Thymic mass was decreased in diabetic mice and restored by the combination therapy. Authors therefore concluded that diabetes might be a bone marrow disease with disturbed thymus and targeting abnormal bone marrow stem cells could a solution to diabetes. Overall, the study sound interesting, and the results will add to the pool of knowledge in the field. However, there are several concerns authors need to address:

Major comments

1. Abstract: Authors need to re-write the abstract and clearly state the aim or hypothesis of the study. The methodology should be described briefly in the abstract.

(Response) We thank the reviewer for he/her comments. We nor rewrite the abstract and clearly describe our hypothesis in this study.

2. Introduction: Authors should rewrite the introduction and update their references on the pathophysiology and management of diabetes and clearly state the justification for this study.

3. Authors should rewrite the whole manuscript for scientific publication

(Response) We now rewrite the whole manuscript and describe the aim and explain the results more clearly.

4. How did authors determine the dose of STZ? How was the dose of insulin and givinostat determined? Why did authors give givinostat every 3 days for 8 weeks? Authors need to state the rationale for choosing the dose, route and duration of administration for reproducibility.

(Response) We used 150mg/kg STZ, which was widely utilized by many investigators to investigate the pathophysiology of diabetes. This dose was also utilized in our previous studies (Katagi et al., Commun Biol 2011; 4:1-14)(Kojima et al, PNAS 2004;101: 2458-2463). In turn, ginivostat was administered every 3 days according to the previous literature (Chan et al., PLoS One. 2013;8(1):e54255). We now add this information as references in the

method section.

5. Did authors determine the wet or dry weight of thymus?

(Response) Thymus weights were evaluated after immersion in 15% sucrose as described in the method of the original version. Since it is likely important information, we now transfer it from extended data to new Fig 4G-H.

6. Why did authors represent their data with SE rather than SD?

We simply prefer to use SE. The following is the figure with SD for the reviewer.

7. Authors should include in their discussion if any, the difference between givinostat and endogenous HDAC inhibitors like short chain fatty acids (butyrate or propionate)? Which one seems superior?

(Response) Unfortunately, we did not examine the effects of endogenous HDAC inhibitors in this study. However, this is an interesting point, so that we will plan to examine it in the future study.

8. Authors stated in the discussion that “the evidence in this study shows that islets that had failed to produce insulin regained insulin expression in response to the combination therapy, which indicates that β -cells retain the ability to regenerate islets” Was the effect of the combination therapy restorative or ameliorative on the islets? This should be made clear in your discussion to guide the readers right.

(Response) This is a very good question, but we are not sure which it is either “restorative” or “ameliorative”, as we did not examine it in detail in the current study. However, we assume that many of islets were regenerated as most of islets which positive for insulin signals were small and number of islets seemed increased in the treated groups compared with control untreated groups. We understand the reviewer’s point but we prefer to refrain from mentioning this issue as we do not have enough data to discuss it. Since it is a very important and intriguing issue, we would like to examine it in the future study.

9. Was there any side effect (s) noticed with BMT?

(Response) although several organs, including lung, bone marrow cells and neurons, are known to be sensitive to radiation, we did not notice any side effects with BMT in this study.

10. Authors should correct all the punctuation, typographical and grammatical errors in the manuscript.

11. The manuscript will also benefit from the service of a professional language editor.

The whole manuscript is now substantially rewritten with careful attention for typographical and grammatical errors with professional language service (Springer Nature Author Services).

Reviewer #2 (Remarks to the Author):

The authors provided a detailed study on how epigenetic modification in bone marrow stem cells is involved in diabetes progression and the regulation of this modification not only stop the progression but also can cure diabetes. The manuscript is well designed and provided the clear insights about the role of BM in diabetes pathogenesis by showing the involvement of normal BM in regulating the normoglycemia phase in glucose-controlled environment. The study also highlights the mechanistic approach of deteriorating insulin secretion from islets by the movement of aberrant BMDCs which state a conclusive remark about the negative impact of abnormal BMDCs migration in β cell regeneration in diabetic condition. The authors proposed removal of abnormal BMDCs as a crucial approach to ameliorate the pathogenesis of diabetes. The authors hypothesized on the ground of differential expression of HDACs in pathogenic BM cells that epigenetic modulation could provide the alternative of impractical BMT to each individual and experimentally provided significant effect of nonspecific HDAC inhibitor along with early glucose control maintained normoglycemia for a period of 8 weeks and carried over to an additional 4 weeks even after inhibitor administration, which is interesting. Interestingly the combination study showed significant regeneration of islet forming β cells. Additionally, the study demonstrated the importance of thymus in regulation of diabetes. The study is well planned and addressed an important aspect which is novel and merits publication.

We request the authors to address following concerns:

a. How authors decided on using givinostat? Is this educated guess, or they have carried out some sort of screening?

(Response) Givinostat was simply selected as this compound is a pan-HDAC inhibitor. We did not test other types of HDAC inhibitors. We now add it in the manuscript

b. Reviewer requests the author to discuss alternative HDAC inhibitor like entinostat or Panobinostat that are reportedly much milder. Especially, Entinostat has been reported by many authors for its role in metabolic disorders. A discussion in this regard will enrich manuscript.

(Response) Thank you for the reviewer's suggestion. While this issue seems important, we feel that it would not be appropriate to mention specific name of HDAC inhibitors. It is because it is not our aim to identify specific type of HDAC inhibitors but to show the effect of

HDAC inhibitors as an alternative of BMT. Givinostat was utilized because this compound is a non-specific pan HDAC inhibitor. Entinostat or Panobinostat is a type-1 HDAC inhibitors, so that we guess it may not be potent enough to cure diabetes in this study.

c. The reviewer would suggest a RNA seq of BMDC with givinostat (if it is not cost prohibitive). Alternatively, a gene expression analysis on the related signaling pathways will enrich the data.

(Response) We thank the reviewer for a key issue in this study. To address the issue, we attempted to isolate RNA from bone marrow tissue fixed with 4% PFA, and synthesize cDNA (SuperScript™ IV First-Strand Synthesis System, ThermoFisher scientific). But unfortunately, this process did not work in our hand. The reason for our failure could be due to degraded RNA in samples. We cannot perform new set of experiments because of financial problem. Another reason is that the new set of experiments also takes more than 6 months to obtain the data.

d. The authors showed 8 weeks of study upon administration of givinostat followed by a four weeks' observation without any treatment. A chronic therapeutic outcome observation should be appropriate to ensure the long-term effect of givinostat upon limited administration as a trend of gradual increase of blood glucose that can be observed after 12th week of treatment (Fig.4b). The chronic study also will be appropriate concerning the deleterious effect of givinostat and a long-term repetitive treatment with givinostat may rise concern.

(Response) We thank the reviewer for comments in this issue. While blood sugar levels show a trend of a gradual increase during observation points, statistical analysis showed that mean glucose concentration in this period was significantly lower and looks as low as the glucose levels in control non-DM mice (shown in Fig 4C). As such, we thought that serum glucose did not increase, but rather fluctuated within normal range. In terms of concerning the side-effect of givinostat, we did not find any side-effects in terms of serum glucose, body weight, and locomotive activity for 4 months after stopping givinostat therapy.

e. The involvement of specific class of HDAC will give more appropriate mechanistic approach and will help in selection of safer and better inhibitor. Having said this, the reviewer noted the novelty of the approach and brilliant execution of the experiments and recommend publication.

(Response) We are pleased to know that this reviewer notes the novelty of this study and recommends publishing this study.

Reviewer #3 (Remarks to the Author):

In their manuscript “HDAC inhibition resets the bone marrow and thymus to cure diabetes”, Kojima and colleagues use bone marrow transplantation experiments to investigate the involvement of the latter in type 1 diabetes mellitus (DM1) pathogenesis. They suggest that BM cells drive DM1 pathology and that HDAC inhibition is sufficient to “cure” the disease. While the authors apparently spent a lot of effort and time on this project, most conclusions in this manuscript are overstated, not supported by experimental data and misleading. This includes the title of the manuscript, which is inappropriate given the experimental design and findings shown. Additionally, the story line is convoluted and hard to follow. In summary, this study is not suitable for publication in the present form and would require extensive editing/revisions.

Please see some major concerns below.

- As mentioned above, several points in this manuscript are overstated. Headings like “Aberrant BM is the cause of diabetes” or statements suggesting having discovered the “cure” for the disease are not appropriate whatsoever, particularly with respect to the data shown. This also holds true for “The thymus is key to curing diabetes” and so on.

(Response) We thank the reviewer for his/her comments. Based on our data, normal glucose concentration was maintained without any therapy for additional 4-weeks, and importantly endogenous insulin production was also recovered during this period. Given these facts, we thought that diabetes was cured. We believe that the importance of thymic function in treating diabetes was clearly demonstrated by the fact that surgical elimination of thymus significantly eliminated the therapeutic effect.

- The authors solely rely on a single animal model that does not recapitulate the true pathogenesis of DM1. Streptozotocin is a toxin directly injuring beta cells, while patients suffering from type 1 diabetes lose beta cells due to autoimmune mediated destruction. If the authors really want to propose having found the cure for a disease, they should use an animal model relevant to the underlying pathology such as NOD mice. This is particularly relevant given the mechanism proposed

(Response) Thank you for pointing this issue out. We need to explain more appropriately on

this point. Basically, in contrast to many investigators who focus on how diabetes is induced or initiated, we are rather interested in a different issue as to why diabetes is not cured after disease onset. Currently, we assume that an interfering mechanism might be operated against β -cell regeneration. This is a common situation regardless of type of diabetes, and we think that we should start from the status of established diabetes. In this regard, STZ model is appropriate because β -cells can be initially destroyed by streptozotocin. NOD mice are a good model to study a causal mechanism for type 1 diabetes, but may not be appropriate to address our issue. We now explain it in detail in new introduction section.

- Throughout the study, authors rely on immunohistochemistry analysis to evaluate insulin and glucagon expression in the pancreas, while circulating insulin or c-peptide levels are never shown. This is a weakness of the study

(Response) We measured serum insulin concentrations, and found that a reduction in serum insulin levels was significantly ameliorated by the combination therapy at 12 weeks in diabetic mice. We now add this data in new Extended data Fig. 3

- The study focuses on contributions of the BM to DM1 pathogenesis and the authors use the term “BMDCs” throughout the manuscript. What are “BMDC cells”? how is this population defined? Why did the authors not narrow this down to a defined cell type?

(Response) The abbreviation of BMDCs stands for “bone marrow-derived cells”, and it was described in the introduction in the original version of this manuscript. In terms of the next question, our group previously narrowed abnormal BMDCs down to “CD106 short-term hematopoietic stem cells, which aberrantly express TNF- α and proinsulin and is likely responsible for the development of diabetic complications. This was explained in the introduction of the original version of this manuscript. We recently published the precise information as following ².

2. Katagi, M. *et al.* Malfunctioning CD106-positive, short-term hematopoietic stem cells trigger diabetic neuropathy in mice by cell fusion. *Commun Biol* **4**, 1–14 (2021).

- How do BMDCs or resident immune cell populations including T-cells respond to streptozotocin? It is unclear whether streptozotocin directly elicits autoimmunity, which would be interesting but irrelevant to DM1 pathogenesis as it is a non-physiological toxin (see previous point)

(Response) We appreciate the reviewer for raising this issue. It is widely believed that streptozotocin predominantly targets β -cells, and to less extent renal proximal tubular cells. However, an old literature suggested that this compound might directly damage bone marrow cells and T cells³. However, this finding has not been intensively followed by researchers so that the details remain unknown. Rather, streptozotocin-induced diabetic models have been currently widely utilized in diabetic research.

3. Nichols, W. K., Vann, L. L. & Spellman, J. B. Streptozotocin effects on T lymphocytes and bone marrow cells. *Clin Exp Immunol* **46**, 627–632 (1981).

- The authors conclude that diabetes is a bone marrow derived disease because they observe that blood glucose levels rise in diabetic mice upon cessation of insulin supplementation following transplantation of bone marrow from streptozotocin-exposed animals. This conclusion is a logical fallacy and not supported by the data shown. To actually prove this, the authors would need to transplant bone marrow from diabetic mice to healthy mice (diabetic \rightarrow non-diabetic) and observe that these animals develop diabetes without any additional intervention. These data are crucial and not included in the manuscript. Even if the authors could show this one should still be cautious classifying diabetes as a “bone marrow” disease, which is an overstatement. Moreover, this would not rule out a direct STZ effect (see previous points)

(Response) We thank the reviewer for raising this important issue. Basically, we do not think that bone marrow transplantation of diabetic animals into wild type mice is capable of inducing diabetes in the recipients. According to a previous literature, healthy islets contain small number of BMDCs (less than 3% of total cells) in the islets⁴. Even if these BMDCs are replaced for abnormal BMDCs, these cells would not be potent enough to cause sufficient damage in the islet. In fact, we performed the experiment that the reviewer indicated, but BMT of diabetic mice failed to induce diabetes in the wild type mice

Here is an additional experiment showing that serum glucose concentration in IPGTT at 8 weeks in nonDM mice receiving bone marrow cells from either DM mice or nonDM mice.

4. lanus, A., Holz, G. G., Theise, N. D. & Hussain, M. A. In vivo derivation of glucose-competent pancreatic endocrine cells from bone marrow without evidence of cell fusion. *J Clin Invest* **111**, 843–850 (2003).

- How do the authors dissect differences between endogenous (beta cell) and exogenous (pellet) insulin production?

(Response) Basically, an insulin pellet lasts for only a couple of weeks, so that after 4 weeks, blood glucose should be controlled by endogenous insulin. We now add this information in the method.

- Immunofluorescence in figure 2 does not look convincing and should be statistically quantified

(Response) We now show the quantified data in new fig2. It clearly demonstrated that BMDCs express Vcam1 and fuse with residential cells in the islets and thymus of diabetic mice.

- “yellow”/fused cells are apparently barely detectable in the pancreas of the mice, which questions their relevance in disease pathogenesis

(Response) In order to make it clear, higher power of views was shown in new Fig.2.

- Overall, it is unclear which relative contributions are attributable to 1.) “cell-fusion”, 2.) direct STZ effects on immune cell populations, 3.) thymic dysfunction, 4.) blood glucose and 5.) defined immune cell populations to the phenotype

(Response) It is true that the mechanism seems complicated, but all components raised by the reviewer are inter-related and cause the pathogenesis of diabetes. Basically, the central player is “abnormal BMDCs”, which can fuse with resident cells in several organs to impair their physiological function. The abnormal BMDCs also cause thymic dysfunction. Since a major component of thymus is BMDCs, thymus function is dysregulated in case that BMDCs are pathologically abnormal. Finally, abnormal BMDCs then disturb the regenerative process in the islets, which then deteriorates glucose control leading to the vicious cycle. This process is illustrated in Fig. 6

- Do BMDCs isolated from diabetic mice express an “autoaggressive” gene signature?

(Response) Yes, those cells express several inflammatory cytokines. Details are shown in Supplemental figure 3 of our previous article (Katagi et al. Commun Biol 2021;4:1-14) We now add this information in the introduction of new version.

- At one point, authors jump from the bone marrow to the thymus, which is confusing to the reader. The thymus is the main tissue for T-cell education, but T-cells are not discussed as relevant players in the first place (which they should). It is also unclear whether “BMDCs” include T-cells (likely yes) and if so, whether depleting T-cells prior to Tx prevents phenotype development

(Response) We don't think that investigating the role of thymus is a jump from bone marrow. It is because thymus is physiologically composed of cells derived from bone marrow so that an alteration in bone marrow stem cells theoretically lead to thymic dysfunction. Having say that, it is kindly presumable that abnormal BMDCs caused the thymic dysfunction which in turn fails to educate abnormal BMDCs. Thus, abnormal BMDCs give rise to dysfunctional T cells, which would play a role in the pathogenesis of diabetes. However, current study did not address the specific role of T cells in the pathogenesis of diabetes.

- If the authors suggest that DM1 is a BM-fueled disease, why would it then be surprising that Tx of wildtype BM is required to eliminate “aberrant BM” cells?

(Response) We do mention that diabetes is a BM-“fused” disease”, but not “a BM-“fueled” disease”.

- The experimental design in Fig. 4 is unclear. Did the authors inject STZ together with givinostat? If so, this would suggest that children with type 1 diabetes would need givinostat therapy even before the onset of diabetes as STZ-induced diabetes takes days to develop

(Response) As we did describe the protocol of this study in the method, we first injected STZ, and then administered either a blank or an insulin pellet before 2.5 mg/kg givinostat (Cayman Chemical, Michigan, USA), an HDAC inhibitor, was administered orally or by injection via the tail vein every three days for eight weeks.”

- The authors found that givinostat monotherapy entailed mortality in WT mice. This is a knock-out criterium for translating these findings into a clinical context.

(Response) We did not have any wild type mice treated with givinostat monotherapy.

- Although the finding that thymectomy blunts the therapeutic effect of givinostat is interesting, it is unclear how this fits within context of the study hypothesis as the thymus and bone marrow harbor distinct immune cell populations (see previous responses)

(Response) As explained in the previous section, we assume that abnormality of bone marrow cells links to the thymic dysfunction because the thymus is composed of bone marrow derived cells. Since curing diabetes was paralleled with the thymic recovery, we hypothesized that thymus plays an important role in curing diabetes. To test our hypothesis, the thymus was surgically removed. We found that surgical removal of the thymus significantly eliminates the therapeutic effects, suggesting that the thymus plays a key role in treating diabetes with this therapy.

- Methods are inappropriately short and do not allow recapitulation of experimental procedures

(Response) We now rewrite methods to provide adequate information of experimental procedures.

- Language editing is recommended

(Response) Language is now edited by professional editors (Nature author service)

Reviewers' comments:

Reviewer #3 (Remarks to the Author):

The revised manuscript is a much-improved version. The reviewer noted the novelty of the concept and recommends the publication of the manuscript.

Reviewer #4 (Remarks to the Author):

The authors have revised their manuscript and implemented several changes. However, many of my suggestions have not been addressed/were ignored. Most importantly, the manuscript remains overstated and the authors' claims remain insufficiently supported by their data.

1. As previously stated, the term "cure" is inappropriate with respect to the experimental data shown. Even if insulin levels are sustained for up to 4 weeks following intervention, this does not prove that diabetes has been "cured". First of all, diabetes is a multi-system disorder, which culminates in organ pathologies such as retinopathy, nephropathy and neuropathy. If the authors want to demonstrate that they "cure" the disease, then they need to show that none of these changes occur during the lifetime of these mice. Patients do not die from low insulin but resulting complications. Second, 4 weeks is an irrelevant short time frame with respect to claim of the authors. This wording needs to be changed.
2. The authors suggest that the streptozotocin model is the most appropriate tool to investigate their hypothesis. Their argumentation is not intuitive and conclusive. B-cell regeneration could certainly also be studied in a mouse model of autoimmune-mediated diabetes mellitus, which, as pointed out previously, would be more relevant to their question. They could also address their question in NOD mice AFTER diabetes has been established so this is not a valid argument either.
3. If the authors cannot rule out that streptozotocin affects T-cell function, then they should look into this experimentally.
4. The BM Tx experiment suggested by myself in the previous review process, in fact, rules out a relevant contribution of the proposed "BMDCs" to beta cell failure and suggest that the author's claim that the bone marrow is responsible for/fuels diabetes pathogenesis is incorrect and overstated
5. The IF quality has improved
6. The mechanism proposed is still very unclear and the complex interactions remain unresolved and not intuitive to follow to the reader
7. "We don't think that investigating the role of thymus is a jump from bone marrow. It is because thymus is physiologically composed of cells derived from bone marrow so that an alteration in bone marrow stem cells theoretically lead to thymic dysfunction. Having say that, it is kindly presumable that abnormal BMDCs caused the thymic dysfunction which in turn fails to educate abnormal BMDCs. Thus, abnormal BMDCs give rise to dysfunctional T cells, which would play a role in the pathogenesis of diabetes. However, current study did not address the specific role of T cells in the pathogenesis of diabetes"

This response nicely illustrates the core problem of this manuscript: it builds on assumptions and premise that is not tested experimentally and thus questions the validity of the study. It also, as pointed out, complicates the interpretation of the results because the readers may not be familiar or "believe in" the same concepts.

8. "We did not have any wild type mice treated with givinostat monotherapy".

Then these mice should be included.

Overall, this manuscript remains superficial, largely overstated and in the present form, not suitable for publication in the proposed journal.

Reviewer #5 (Remarks to the Author):

I think the authors have addressed the issues raised by reviewers in the main. However, they have not addressed R#3's comments regarding over-statement of their findings. I do take the authors' point that the experimental data support their point. However, the data were obtained from a mouse model and this should be properly acknowledged in the title and in the main body of the manuscript for clarity. This is needed as there are differences in terms of beta cell regeneration between human and mouse.

Reviewer #4 (Remarks to the Author):

The authors have revised their manuscript and implemented several changes. However, many of my suggestions have not been addressed/were ignored. Most importantly, the manuscript remains overstated and the authors' claims remain insufficiently supported by their data.

1. As previously stated, the term "cure" is inappropriate with respect to the experimental data shown. Even if insulin levels are sustained for up to 4 weeks following intervention, this does not prove that diabetes has been "cured". First of all, diabetes is a multi-system disorder, which culminates in organ pathologies such as retinopathy, nephropathy and neuropathy. If the authors want to demonstrate that they "cure" the disease, then they need to show that none of these changes occur during the lifetime of these mice. Patients do not die from low insulin but resulting complications. Second, 4 weeks is an irrelevant short time frame with respect to claim of the authors. This wording needs to be changed.

(Response) We agree with the comments from reviewers and editors and now change the title of our manuscript to be more specific and to emphasize that this study focuses on the mouse model of streptozotocin-induced diabetes. In addition, the word of "cure" is now not used in the whole manuscript to explain our results.

New title is the following;

"Complete remission with a transient stepwise therapy using an HDAC inhibitor and insulin in streptozotocin diabetic mice"

2. The authors suggest that the streptozotocin model is the most appropriate tool to investigate their hypothesis. Their argumentation is not intuitive and conclusive. B-cell regeneration could certainly also be studied in a mouse model of autoimmune-mediated diabetes mellitus, which, as pointed out previously, would be more relevant to their question. They could also address their question in NOD mice AFTER diabetes has been established so this is not a valid argument either.

(Response) We now change the title to focus on the streptozotocin-induced diabetic mice in this manuscript. We will study NOD mice in the future study.

3. If the authors cannot rule out that streptozotocin affects T-cell function, then they should look into this experimentally.

(Response) We thank the reviewer for raising this issue. We performed FACS analysis of total thymic cells from control non-diabetic mice and streptozotocin induced diabetic mice, and found that there was no difference in the percentage of total T cell number in the total thymic cells between control non-diabetic mice and streptozotocin-induced diabetic mice ($10.8 \pm 1.8\%$ vs. $9.9 \pm 0.9\%$, respectively). Dot plot diagrams showed that the fractions of CD4+ and CD8+ cells in total CD3+ cells in control non-diabetic mice are also similar to those in STZ-induced diabetic mice (Supplementary Figure 2a). Likewise, immunohistochemistry showed that the number of CD8a in the thymus of non-diabetic mice were not different from those in diabetic mice (Supplementary Figure 2b). In addition, Fig 4d shows that insulin therapy prevented thymic atrophy in streptozotocin-induced diabetic mice. Those data suggest that streptozotocin would not have significant effects on the T cell functions in the thymus.

4. The BM Tx experiment suggested by myself in the previous review process, in fact, rules out a relevant contribution of the proposed “BMDCs” to beta cell failure and suggest that the author’s claim that the bone marrow is responsible for/fuels diabetes pathogenesis is incorrect and overstated

(Response) Thank you for the comments. We believe this issue could be addressed by experiment 1, in which we utilized bone marrow transplantation to examine the role of BM cells in streptozotocin-diabetic mice. As shown in Fig 1., eliminating bone marrow cells from diabetic mice succeeded to improve glucose levels and recover insulin expression in the islets. These data suggest that bone marrow cells in diabetic mice contribute to the islet dysfunction in diabetic mice.

5. The IF quality has improved.

(Response) we are glad to hear that the reviewer thinks that the quality of our data is OK.

6. The mechanism proposed is still very unclear and the complex interactions remain unresolved and not intuitive to follow to the reader

(Response) We thank the reviewer for raising this issue. We now simplify to show the mechanism in new Fig. 6, which we hope to illustrate that high glucose initiates BM dysfunction, resulting in both pancreatic and thymic dysfunctions. Pancreatic dysfunctions sustain hyperglycemia. In turn, thymic dysfunction could lead to pancreatic dysfunction based on our results showing that the surgical elimination of thymus blocked pancreatic recovery in response to the treatment. While the mechanism remains to be determined, recent studies demonstrated that the regulatory T cells are required for the tissue repair (Burzyn D., Cell 2013; 155:1282-2195)(Dominguez-Villar M., Nat Immunol 2018;19:665-673), indicating an indispensable role of the thymus for recovery of the pancreas from diabetic insults. In order to block the vicious cycle, both glucose control and an HDAC inhibition are required. Blood controls could eliminate the future production of abnormal BM cells whereas an HDAC inhibition would remove the diabetic memory from existing deleterious BM cells.

7. “We don’t think that investigating the role of thymus is a jump from bone marrow. It is because thymus is physiologically composed of cells derived from bone marrow so that an alteration in bone marrow stem cells theoretically lead to thymic dysfunction. Having say that, it is kindly presumable that abnormal BMDCs caused the thymic dysfunction which in turn fails to educate abnormal BMDCs. Thus, abnormal BMDCs give rise to dysfunctional T cells, which would play a role in the pathogenesis of diabetes. However, current study did not address the specific role of T cells in the pathogenesis of diabetes”

This response nicely illustrates the core problem of this manuscript: it builds on assumptions and premise that is not tested experimentally and thus questions the validity of the study. It also, as pointed out, complicates the interpretation of the results because the readers may not be familiar or “believe in” the same concepts.

(Response) We appreciate the reviewer's comments and agree that we should explain the reason why we examined thymus in this study. Given that thymus is physiologically composed of bone marrow cells while bone marrow disease is closely associated with thymic dysfunction in autoimmune disorder (Nakamura T., *Thymus* 1985;7:151-60), we simply hypothesized that "an alteration in bone marrow stem cells theoretically lead to thymic dysfunction in the streptozotocin-induced diabetic mice". We found that the thymus in diabetic mouse were atrophic while the combination therapy suppressed the pathological alterations. We then investigated if the atrophic thymus might be involved in the therapeutic process. Interestingly, the surgical elimination of the thymus blocked the protective effects of the combination therapy, suggesting a key role of thymus in treating the streptozotocin-induced diabetes.

8. "We did not have any wild type mice treated with givinostat monotherapy". Then these mice should be included.

(Response) In the previous comment, the reviewer mentioned to clarify if givinostat monotherapy entailed mortality in WT mice. Regarding this issue, there is a previous study examining the mortality of wild type mice treated with givinostat. In that study, they administered 5mg/kg and 10mg/kg of givinostat to wild type mice daily for three months (Regna NL., *Clin Immunol* 2014;151:29-42), and found that those treated mice were alive through the end of study and no mice were dead. In turn, we utilized much lower dose of givinostat, which was 2.5mg/kg every three days for 8 weeks, and therefore, we would guess that the dose of givinostat in our protocol would not kill any wild type mice.

REVIEWERS' COMMENTS:

Reviewer #4 (Remarks to the Author):

The manuscript has been toned town but the data presented still does not adequately support the claimed findings.

Reviewer #4 (Remarks to the Author):

The manuscript has been toned town but the data presented still does not adequately support the claimed findings.

(Response)

Thanks to several critiques from Reviewer 4 during several times of revision process, we believe that our manuscript was substantially improved. This time, we carefully went through all sentence and tried to accurately describe each finding, which are now highlighted with red in this manuscript. Some sentences were re-written in the section of results and several sentences were removed or modified in the discussion.